# Efficacy of Endodontic Disinfection Protocols in an *E. faecalis* Biofilm Model—Using DAPI Staining and SEM

**DOI:** 10.3390/jfb14040176

**Published:** 2023-03-23

**Authors:** Maria Dede, Sabine Basche, Jörg Neunzehn, Martin Dannemann, Christian Hannig, Marie-Theres Kühne

**Affiliations:** 1Policlinic of Operative and Pediatric Dentistry, Medical Faculty Carl Gustav Carus, TU Dresden, 01307 Dresden, Germany; 2Department of Endodontics, School of Dentistry, National and Kapodistrian University of Athens, 11527 Athens, Greece; 3Vertriebsgesellschaft GmbH, Geistlich Biomaterials, Schneidweg 5, 76534 Baden-Baden, Germany; 4Faculty of Automotive Engineering, Institute of Energy and Transport Engineering, Westsächsische Hochschule Zwickau, 08056 Zwickau, Germany

**Keywords:** bacterial penetration, biofilm model, DAPI method, dentinal tubules, *Enterococcus faecalis*, root canal irrigants

## Abstract

The aim of this study was to investigate the antimicrobial efficacy of different disinfection protocols in a novel *Enterococcus faecalis* biofilm model based on a visualization method and to evaluate the potential alteration of dentinal surface. A total of 120 extracted human premolars were allocated to 6 groups with different irrigation protocols. The assessment of the effectiveness of each protocol and the alteration of dentinal surface were visualized by using SEM and fluorescence microscopy (DAPI). A dense *E. faecalis* biofilm with a penetration depth of 289 μm (medial part of the root canal) and 93 μm (apical part) validated that the biofilm model had been successfully implemented. A significant difference between the 3% NaOCl groups and all the other groups in both observed parts of the root canal (*p* < 0.05) was detected. However, the SEM analysis revealed that the dentinal surface in the 3% NaOCl groups was severely altered. The established biofilm model and the visualization method based on DAPI are appropriate for bacterial quantification and evaluation of the depth effect of different disinfection protocols in the root canal system. The combination of 3% NaOCl with 20% EDTA or MTAD with PUI allows the decontamination of deeper dentine zones within the root canal but simultaneously alters the dentinal surface.

## 1. Introduction

The goal of an endodontic treatment should be the elimination of microorganisms and the prevention of a possible reinfection. A successful root canal therapy relies on the combination of proper instrumentation, irrigation, and obturation of the root canal system. The etiology behind endodontic treatment failures is mainly a persisting infection with a biofilm structure in the root canal system [1]. Unfortunately, the root canal system with its anatomical complexity represents a challenging environment for the effective removal of bacteria and biofilm [2]. A plethora of chemical irrigants activated by different technical devices are used to eliminate residual microbes in root canals [2,3]. One of the most common techniques is passive ultrasonic irrigation (PUI) [4,5,6].

A microorganism that has been intimately associated with treatment failures is *Enterococcus faecalis* (*E. faecalis*) [7,8]. The reported prevalence of *E. faecalis* ranges from 24% to 77% in post-treatment root canal infections [9]. To date, it is still not possible to explain this prevalence since the origin of *E. faecalis* infections remains unknown. Since its first description in 1906, it is termed *Streptococcus faecalis* or “Streptococcus of faecal origin” as it has often been recovered from fecal matter or sewage [10]. It is both a commensal pathogen of the gastro-intestinal tract and a common nosocomial pathogen. Its transition from commensal to pathogenic is far from being completely understood. *E. faecalis* is most likely not derived from the endogenous flora or from nosocomial transmission but is instead a food-borne pathogen in root canal infections [11]. However, *E. faecalis* is considered a suitable model for studying bacterial infections in root canals [12], specifically in in vitro studies.

Extensive research has been accomplished in the field with regard to bacterial reduction of biofilm within the root canal system [13,14]. However, there are limited studies that compared the synergistic effects of passive ultrasonic irrigation with different irrigants against *E. faecalis* biofilm in the root canal system, as well as their effects on the dentinal root canal surface. The use of passive ultrasonic irrigation has been limited to endodontic irrigants, such as NaOCl and EDTA, and its use over CHX or MTAD (mixture of tetracycline, acid, and detergent), whereas the combination of different irrigants has not been studied in detail [6,15].

Microscopic techniques have been used for the evaluation of the effects of various endodontic irrigants on biofilms [16,17]. The dentinal root canal surface was evaluated by scanning electron microscopy (SEM) in this study. Fluorescence staining with DAPI—a fluorescent dye to visualize bacteria by binding to the AT-rich regions of nucleic acids of double-stranded DNA, thereby forming fluorescent units—was used to detect bacteria in the depth of dentinal tubules [18,19]. In the present study, the degradation and removal effects of different disinfection protocols were investigated using a visualization method. The aim of our study was to establish a biofilm root canal model in order to visualize and quantify bacterial colonization within the dentinal tubules after the application of different disinfection protocols and to identify the synergistic effects of the different irrigants in combination with PUI on the dentinal surface for the first time.

## 2. Materials and Methods

### 2.1. Sample Preparation

A total of 120 extracted human single-rooted premolar teeth were used in this study. The anatomic crown of each tooth was resected horizontally at 16 mm with a diamond disk (Diamond Disc 330 CA, Struers, Willich, Germany). The working length of each root canal was measured at 15 mm. The roots were prepared using the file F1 (020/06) of the rotary endodontic nickel–titanium system, ProTaper Universal (Dentsply-Maillefer, Switzerland), and were irrigated with 2 mL of NaCl (0.9%). Finally, the root canals were rinsed with 2 mL of 20% EDTA for 1 min under agitation with the ultrasonic tip IRRI S/25 mm (Satelec-VDW GmbH, Munich, Germany) for the removal of the smear layer and again with 0.9% NaCl. In order to prepare all samples for the splitting of the roots at the end of the procedure, two lines were drawn longitudinally on the buccal and lingual planes of each root. Longitudinal grooves were then carved with a diamond bur under the caution of not invading the root canal along the drawn lines (Figure 1). The complete longitudinal fracture of the roots was performed with a razorblade (Herkenrath, Solingen, Germany) and a hammer (Braun, Tuttlingen, Germany), providing two root halves from each sample.

Lastly, the teeth were cleaned in an ultrasonic bath (Sonorex Digital 10P, Bandelin, Berlin, Germany) with 20% EDTA (10 min) and finally in distilled water (1 h) (Aqua destillata, Weinert Wassertechnik GmbH, Dresden, Germany). The apical foramen of each root and the lateral grooves were sealed with silicone (Provil novo, Heraeus Kulzer GmbH, Germany) to avoid bacterial leakage through the apex and the lateral canals or through the dentinal tubules during the procedure of inoculation with *E. faecalis*. Afterward, the teeth were placed in an ultrasonic bath for 10 min with tryptic soy broth (TSB—Merck, Darmstadt, Germany), followed by autoclaving (10 min, 121 °C). To check the sterility of the samples, the teeth were incubated in the TSB for three days at 37 °C to prove that no contamination, as indicated by the cloudiness of the TSB, took place. Afterward, the teeth were embedded in a 5 mL Eppendorf tube (Eppendorf AG, Hamburg, Germany) with 3% Agarose (Merck, Darmstadt, Germany). Again, each root was filled with the TSB medium using a fine 27-gauge needle (3/4 0.4 mm × 19 mm) (Transcodent, Germany) to keep the dentine moistened.

### 2.2. Inoculation of the Roots with E. faecalis and Incubation

The *E. faecalis* strain was obtained (clinical bacterium isolated from a patient with persistent root canal infection and approved by the ethics committee of the University of Freiburg 140/09) and cultivated in a S2 laboratory. A day prior to the inoculation procedure of the roots, a tryptic-soy-broth (TSB) bouillon, including 2 mg/mL of Streptomycin and 0.2 μg/mL of Amphotericin B, was inoculated with a single *E. faecalis* colony on an agar plate and incubated at 37 °C. Streptomycin (2 mg/mL) was added as an antibacterial agent to the TSB in order to inhibit the growth of other bacteria and Amphotericin B was added as an antimycotic agent. On the first day of the inoculation procedure, each root in the Eppendorf tube was inoculated with 77*E. faecalis* culture bouillon (1.6 × 10^8^/mL) after the removal of the old TSB medium and incubated at 37 °C. The next day, the bacterial suspension within the root canal was again removed with a 27-gauge needle, and new *E. faecalis* culture bouillon was applied to the roots and incubated at 37 °C. After two days, the bacterial suspension was removed, but this time, it was not renewed; instead, fresh TSB medium was applied to the roots. Furthermore, the infected root canals underwent a renewal of the TSB medium every day for 6 weeks. The incubation time lasted 6 weeks to allow the formation of *E. faecalis* biofilm.

### 2.3. Application of the Disinfection Protocols

After the incubation period, the bacterial suspension was removed from all the roots, and the samples were randomly divided into five experimental groups (*n* = 20) and one control group (*n* = 20). Group 1 (CTR): control group with no disinfection protocol; Group 2 (NEC_PUI_): NaOCl 3% + EDTA 20% + CHX 2% + NaCl 0.9% under passive ultrasonic irrigation (PUI) with Irri S 25 (Satelec-VDW GmbH, Munich, Germany); Group 3 (NE_PUI_): NaOCl 3% + EDTA 20% under PUI with Irri S 25; Group 4 (CE_PUI_): CHX 2% + EDTA 20% under PUI with Irri S 25; Group 5 (NM_PUI_): NaOCl 3% + MTAD under PUI with Irri S25; and Group 6 (NaCL_PUI_): NaCl 0.9% under PUI with Irri S 25 (Table 1). Before the disinfection procedure of any group (including the control group), the roots of all groups were instrumented up to file F4 (040/0.6) of the ProTaper system (Dentsply-Maillefer, Ballaigues, Switzerland) and were rinsed intermittently with 0.9% NaCl solution.

### 2.4. Preparation of the Specimens for Visualization Techniques

The splitting of the roots was performed directly after the application of each disinfection protocol, providing two halves of each sample. One-half of the root was used for the analysis by scanning electron microscopy (SEM), and the other half was used for the analysis by fluorescence microscopic staining with 4′,6-Diamindin-2-phenylindole (DAPI). The halves for DAPI staining were transferred directly into a 15 mL centrifugal tube (Sarstedt, Nürnberg, Germany) with 4% formaldehyde to fixate the bacteria. The fixation lasted 48 h at 4 °C. Subsequently, the roots were placed in Osteosoft^®^ (Merck, Darmstadt, Germany)—for the decalcification of the dentinal roots—until the specimens were sliceable with a scalpel. Then, every root canal half was cut transversally into two pieces: the medial part and the apical part (1 mm before the apex). Shortly after, the embedding of the medial and apical root pieces in paraffin was carried out. The sectioning of the root pieces into 2 μm sections with a microtome (Leica Biosystems Nussloch GmbH, Nußloch, Germany) took place. The object carrier was silanized, and the test species were placed on glass on a histological slide. These final steps were performed for visualization with the DAPI method.

### 2.5. DAPI

DAPI staining (Merck, Darmstadt, Germany) was conducted as described in previous research [20,21]. DAPI (4′,6-diamidino-2-phenylindole) stains DNA unspecifically by binding to the AT-rich regions of double-stranded DNA. The following steps were used for the DAPI staining. The test species were covered with the DAPI stock solution (1.5 μL of stock solution in 500 μL of PBS (phosphate-buffered saline—Invitrogen Ltd., Bend, OR, USA)) in a dark chamber. This DAPI solution was removed after 15 min by rinsing several times with the PBS (phosphate-buffered saline, Invitrogen Ltd., Bend, OR, USA) before the samples underwent fluorescence microscopic analysis [19]. Thereafter, the samples were dried at room temperature and coated with the Vectrashield mounting medium (Sigma-Aldrich, Taufkirchen, Germany). The analysis by epifluorescence microscopy (Axioplan, Zeiss, Oberkochen, Germany) was conducted. The root canal samples with the dentinal tubules were analyzed at 1000-fold, 400-fold, and 100-fold magnifications using a light filter for DAPI (BP 365, FT 395, LP 397 Zeiss, Oberkochen, Germany). The area of the ocular grid allowed the visualization of the total length of the dentinal tubules.

### 2.6. Scanning Electron Microscopy

Regarding the scanning electron microscopic (SEM) investigation, the other half of the root was used. The sectioned root canal specimens with four roots from each group were transferred into microtubes with 4% glutaraldehyde (Sigma-Aldrich, Taufkirchen, Germany). The fixation with glutaraldehyde was continued for 2.5 h at room temperature. The next step was washing with the PBS for 15 min twice and dehydration with Isopropanol (Carl Roth GmbH Co. KG, Karlsruhe, Germany). Then, chemical drying through iterative transfer into hexamethyldisilazane (HMDS) was performed. The specimens were fixed on SEM stubs and sputtered with gold–palladium. The scanning electron microscopy took place using a Philips ESEM XL 30 in the high-vacuum mode to detect secondary electrons for imaging.

### 2.7. DATA Evaluation

First, the surface area of each sample was calculated, using the Image J2-Fiji program (Curtis Rueden of UW-Madison LOCI, Madison, WI, USA), in the DAPI images with a 100-fold magnification [22]. Then, the number of bacteria per surface area was calculated visually by two operators using a compact manual cell counter (Fisherbrand^TM^, Schwerte, Germany) in all the DAPI images of each group (apical and medial parts of the root canal). The arithmetic estimation of bacterial penetration depth was performed using the Axio Vision program (Zeiss, Jena, Germany). By that means, the distance between the entrance of the dentinal tubules and the penetrated bacterial cells was measured. Moreover, all cells were counted between the entrance of the dentinal tubules and the deepest penetrated cell in the specimens (Figure 1).

A four-point scoring system was adapted to evaluate the surface profile of the root canal dentine after the application of the disinfection protocols by using the SEM data [23]. The scoring system was defined according to the representative images from the SEM data (Figure 2):Score 0: Absence of irregularities and dentinal tubules closed.Score 1: Partially irregular and dentinal tubules partially opened.Score 2: Damage of the surface and dentinal tubules opened.Score 3: Severe erosion of the dentinal surface and dentinal tubules widely opened.

### 2.8. Statistical Analysis

The values were compared by using one-way analysis of variance (ANOVA), followed by a post hoc test (Dunnett’sT3). The Dunnett’s T3 test was used to assess the differences between the six groups based on the DAPI data. The level of significance was set at 0.001 for the one-way ANOVA test, with a statistical power of 95%, and at 0.05 for the Dunnett’s T3 test, with a statistical power of 80%.

## 3. Results

The bacterial colonization in the samples was successful. After six weeks of incubation with *E. faecalis*, the cells had already migrated into the dentinal tubules. The examination with the DAPI method gave insight into the remaining bacterial cells inside the dentinal tubules after the application of the different disinfection protocols. All cells were counted between the entrance of the dentinal tubules and the deepest penetrated cell in the specimens. The overall penetration depths of the deepest remaining *E. faecalis* cells in all groups and both parts of the root canal (medial and apical) were measured to validate the biofilm model. A deeper penetration pattern of *E. faecalis* into the dentinal tubules was observed in the medial part of the root canal compared to the apical part of the root canal in the control group (Figure 3).

Bacterial cells were traceable in all specimens by the DAPI method in the control group. The root canals inoculated with *E. faecalis* were heavily infected, and microorganisms were observed in all areas of the dentinal tubules in the control group, even after the instrumentation up to ProTaper file F4 (040/0.06). In general, not only a deeper penetration of *E. faecalis* was observed in the medial part compared to the apical part of the root canal in each group, but the bacterial count was higher as well. More specifically, in the control group, the bacterial count in the medial part was 1492.0 ± 768.4 bact./μm^2^, and in the apical part of the root canal, the count was 172.3 ± 222.1 bact./μm^2^. The representative images from each group visualize the penetration depth of the deepest remaining bacterial cells, as well as the residual infection of the dentinal tubules with the remaining bacteria in general, after the application of the different disinfection protocols. These images from the DAPI data are representative examples of each group from both parts of the root canal (medial and apical) (Figure 4a,b).

Specifically, less bacteria were detected in Group 2 (NaOCl 3% + EDTA 20% + NaCl 0.9% + CHX 2% with PUI) when compared to Group 4 (CHX 2% + EDTA 20% with PUI) and Group 6 (NaCl 0.9% with PUI). These Groups—4 and 6—yielded comparable amounts of bacteria. Hardly any bacteria were detected in Group 3 (NaOCl 3% + EDTA 20% with PUI) and in Group *5* (NaOCl 3% + MTAD with PUI) after the application of the disinfection protocols. The post hoc comparisons using the Dunnett’s T3 test indicated that the mean score for the control group was significantly different from all the other tested groups regarding the medial part of the root canal. However, at the apical part of the root canal, the control group did not significantly differ from Groups 2 (NaOCl 3% + EDTA 20% + NaCl 0.9% + CHX 2% with PUI), 4 (CHX 2% + EDTA 20% with PUI), and 6 (NaCl 0.9% with PUI). There was also no significant difference between Groups 3, 5, and 2 at the medial/apical part of the root canal, but the difference was statistically significant between Groups (3 and 5) and (1, 4, and 6) at the medial part. No statistically significant association could be found between Group 3 and Group 5 (Figure 5).

The SEM data confirmed the results of the DAPI analysis. The visualization of bacteria under the scanning electron microscope (SEM) indicated colonies of *E. faecalis*, especially at the entrances of the dentinal tubules and the root canal surface. It was mainly observed by the SEM how each disinfection protocol had affected the structure of the root dentine.

A considerable alteration of the dentinal structure in the root canal was observed in Groups 3 and 5. This alteration resulted in the erosion of the dentinal ultrastructure. Partial irregularities were observed in Groups 2 and 4. A morphological change of the dentine surface was observed by the SEM in Group 6, as well. The dentinal tubules were partially opened, and there were areas of the dentine surface that were apparently mechanically prepared and instrumented due to the use of the ultrasonic tip (Figure 6).

## 4. Discussion

With the intention of studying the effect of different disinfection protocols, it was necessary to analyze the cleansing effect of different irrigations and to quantify the remaining bacteria after the decontamination. Therefore, a mono-species biofilm model was established in order to quantify penetrated bacteria into the dentinal tubules of human root dentine after the application of different disinfection protocols. The composition and structure of the endodontic biofilms are highly variable. Although this study was not in vivo and did not mimic complex multispecies endodontic biofilms, the established mono-species *E. faecalis* isolated from a biofilm model with persistent root canal infection provided a well-standardized anatomical and biologically relevant model that allowed the comparison of different disinfection protocols against *E. faecalis* biofilms, as visualized with the DAPI method.

The analytical procedure based on the visualization by DAPI staining was established by the present research group [24,25] to quantify the remaining penetrated *E. faecalis* cells within the dentinal tubules and to evaluate the disinfection effect of five different irrigation protocols. To a certain extent, this technique permits the two-dimensional imaging of bacteria on the root canal surface and inside the dentinal tubules. The examination following DAPI staining gave an additional insight into the remaining penetrated bacterial cells into the dentinal tubules [24,25] and confirmed the penetration of bacteria into the dentinal tubules. Furthermore, the examination with the DAPI method revealed a semi-quantitative assessment of bacteria in the observed areas. Undeniably, the evaluation of the results, specifically the calculation of bacteria, was time consuming. The compact manual cell counter allowed us to register each individual colony prior to testing based on standardized comparisons [26]. Notwithstanding the fact that the method was time-consuming, it provided an accurate estimation of the number of bacterial cells.

A criticism often brought up in relation to all imaging techniques used to evaluate intraradicular biofilms is that the observed areas are to some extent subjectively chosen by the examiner. Using the DAPI method, which also provides a local estimation of remaining bacteria, the cross sections of the root—and not of the whole root canal—were evaluated. Certainly, the field of observation is more extended in contrast to other methods, such as colony-forming units (CFU). Considering that the field of vision under the microscope is limited, most of bacteria remain undetected. Regarding CFU, most of the studies are based on the paper point sampling process for the analysis of bacteria, which is considered controversial since it leads to a biased collection of biofilm material that are readily accessible, while hidden bacteria in dentinal tubules are oftentimes disregarded [27]. A negative culture result by CFU does not necessarily imply a bacteria-free root canal system, as bacteria may be retained in complex areas of the system or into the dentinal tubules embedded within a biofilm, thus being inaccessible to the paper points used for sampling. Therefore, the CFU method in combination with paper points for bacterial identification can result in an underestimation of the bacteria present in an infected root canal [28]. Even though the field of vision in the presented DAPI method in this study is limited, it offers insight into the bacterial contamination within the dentinal tubules and the bacterial penetration depth. Further analysis of the root canal wall using SEM gives additional insight into the bacterial contamination of the root canal surface as well as the alteration of the root dentine caused by endodontic irrigants.

It is apparent that the combination of different microscopic techniques is more likely to facilitate a deeper and more realistic analysis of biofilm architecture in the root canal. The combination with SEM provides more information regarding the visualization of bacteria [29]. SEM also provides information concerning the condition of bacteria and the ultrastructure [30,31]. SEM has been used to visualize the distribution of bacteria on the surface of biofilm in the root canal wall, as well as the penetration inside dentinal tubules [24,32,33]. However, the resulting images are, therefore, only pseudo three-dimensional. As biofilms are multileveled, SEM is unable to assess the full depth of these structures [29,34,35]. Another aspect regarding SEM is that only topographical assessment of the observed structures is possible, which makes it nearly impossible to quantify the bacteria into the root dentine areas and, especially, inside the dentinal tubules. Therefore, only qualitative assessment of observed specimens can be performed with this technique. This is not surprising, considering the field of vision under a scanning electron microscope contains only a few micrograms of dentine. For this reason, previous studies have used SEM in order to visualize bacteria in the root canal and not to quantify them [12,29,36]. However, SEM is a very effective method to analyze ultrastructural surface alterations after an irrigant application. Different studies have already shown the use of SEM to investigate enamel [37,38] and dentine [39,40]. Nevertheless, a direct comparison of irrigant decontamination efficacy using fluorescence microscopy (DAPI) and dentinal surface alteration using SEM was performed for the first time in the present study.

Regarding the viability of the remaining bacteria, the established DAPI method is unable to provide knowledge about the viability of such bacteria. There is no evidence regarding a live/dead staining with the advantages of the DAPI method, which has the ability to detect the viability of penetrated bacteria located within the dentinal tubules of the root canal. The fixation of bacteria—a necessary step in the analysis with the DAPI method—destroys the viability state of cells. In general, it can be assumed that the differentiation of viable and dead bacteria is possible using different live/dead staining methods. These methods represent the viability state during the staining procedure. However, with additional after dye accumulation inside the cells, bacteria, indeed, lose their viability [41].

The model in this study was based on a mono-species biofilm with *E. faecalis* within the root canals and dentinal tubules. *E. faecalis* plays an important role in bacterial biofilm invasion and is considered a suitable model for assessing root canal bacterial penetration. Many in vitro investigations have been undertaken to examine the mechanisms involved in bacterial penetration into dentine and to visualize this infection in dentinal tubules [12,42]. Confocal laser scanning microscopy (CLSM) and the DAPI method—which was used in this study—allow the most precise assessment of bacterial penetration in the dentinal tubules and generate less risk of creating artifacts, when compared to SEM [43]. Most of studies with CLSM use colony-forming units (CFU) in order to quantify bacteria that have invaded into dentinal tubules. Considering the drawbacks of CFU, the DAPI method can be considered as an alternative visualization method for quantifying bacterial penetration.

In this study, the effectiveness of the five different disinfection protocols was examined in comparison to the control group, and it was concluded that the disinfection protocols in Groups 3 (NaOCl 3% + EDTA 20% with PUI) and 5 (NaOCl 3% + MTAD with PUI) were the most effective (*p* < 0.001). There was a significant difference in the bacterial count between both Groups 3 and 5 and all the other groups in the medial part of the root canal (*p* < 0.05). Although PUI improved the effectiveness of conventional irrigation, no significant difference was detected between the control group (only instrumentation and conventional irrigation with NaCl 0.9%) and Group 6 (instrumentation and passive ultrasonic irrigation with NaCl 0.9%) based on the Dunnett’s T3 test (*p* < 0.05). Obviously, the irrigants play an important role in the decontamination of *E. faecalis* biofilm, but the use of passive ultrasonic irrigation also enhances bacterial reduction from the root canal systems when compared to other methods of irrigant activation and conventional syringe irrigation [6,44,45,46].

Both groups (Groups 3 and 5) revealed the best results concerning decontamination and biofilm dissolution capacity. NaOCl is appropriate as an irrigant because it is effective in disrupting biofilm [47]. However, structural deformations/alterations were observed in the dentine ultrastructure. The use of NaOCl in combination with additional irrigants as the final irrigation provokes severe structural changes in the dentinal collagen. These phenomena has also been observed in previous ultramorphological studies [48,49]. The erosion of the dentine by proteolytic degradation is followed by the formation of fragile, spongy-like root dentine. The worst-case scenario is root fracture due to the weakening of the root. Therefore, it is a fine line between removing too much dental tissue, which would strongly weaken the root, and leaving the infected dental tissue in the root canal, which would reduce the possibility of achieving the best decontamination effects.

Intensive research is being conducted to develop disinfection protocols for the root canal system. The ideal protocol of disinfection does not exist yet. In the near future, nanotechnology might be applied to the endodontic field. In endodontics, there are no techniques that promote total anti-biofilm removal while simultaneously do not affect the root dentine ultrastructure. Nanomaterials and nanocarriers could open new opportunities for the removal of biofilms and repair of tooth structure [50].

## 5. Conclusions

In conclusion, a clinically relevant *E. faecalis* biofilm model for in vitro studies—based on visualization by DAPI staining—was established. The results of the present study suggest that the combination of 3% NaOCl under passive ultrasonic irrigation with an additional final irrigant, such as 20% EDTA or MTAD, is the most effective disinfection protocol against *E. faecalis* biofilm. Yet, the dentinal root canal surface is altered the most after the application of 3% NaOCl with 20% EDTA or MTAD. The combination of epifluorescence microscopy with DAPI staining and scanning electron microscopy (SEM) is a novel approach of the present research group to visualize and quantify the decontamination effects of different endodontic irrigants by evaluating the remaining bacteria on the root canal surface and within the dentinal tubules. At the same time, the penetration depth of bacteria and the bacterial ultrastructure/condition, as well as the alteration of the dentine due to the endodontic irrigants, can be evaluated.

## Data Availability

The data presented in this study are available from the corresponding author upon request.

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
