# Peer review of "Efficacy of Endodontic Disinfection Protocols in an E. faecalis Biofilm Model—Using DAPI Staining and SEM"

_jfb, 2023, doi:10.3390/jfb14040176_

Round 1

Reviewer 1 Report

The aim of this study  is to aim of this study was to investigate the antimicrobial efficacy of disinfection protocols in novel Enterococcus faecalis biofilm model based on a visualization method and to evaluate the potential alteration of the dentinal surface. This research is under the scope of this journal; the topic is relevant for readers, and this research deals with potentially significant knowledge to the field. And It will be important of  Endodontics knowledge. The topic is relevant for readers and this study deals with potentially significant knowledge to the field and open new way for future studies. 

However, there are some aspects which is need to be improved in the manuscript:

- Correct typos in all manuscript.

  • The use of personal pronouns should be avoided. Example “We have…etc”

(Keywords)

  • Please add keywords, and order the keywords / Mesh terms alphabetically without numbers.

(Introduction) 

What is the importance of this study? What is the gap in this field of literature?

    • You do not think this study is included in the others already done? Which results are comparable with other studies? What has this study been new ?
    • Identified the aim and  null hypothesis on the end of the introduction.

M&M

- It is important to have names in the groups, however the denomination of 1, 2, 3… to identified the the groups it’s sometimes confuse.  Please standardised the name of the Groups: my suggestion used the name of the  materials.

  • This section would better communicate to readers if restructured. A flow chart or diagram of the experimental processing would be valuable.
  • How was the sample calculated? Did the authors perform a power analysis to evaluate if this sample size was appropriate?
  • When mentioning materials or devices: for some of them you don't mention the manufacturer at all, for some you mention only the manufacturer, for some the manufacturer and city, for some you mention the manufacturer and city/ country.

  • How many operators performed the Experimental Study? And how many time you repeat the experimental study?

(Results) 

  • Improve the resolution quality of all figures and graphs (and a presentation). The font/language in the figure/caption is different from the text. Please, standardise the size and the font in the figures and charts with the font of the manuscript. 

(Discussion)

  • Please, identified what was the strength(s) and more limitations of this study? And  read https://doi.org/10.1016/j.addr.2023.114731 for the  implications for future perspectives.

References

  • The titles of references have a different format, 
    the title of the article is written in capital letters at the beginning of words, others only in lower case. Also, the standardized format of presentation in the journal's name. Because names have written in a different format, one is not abbreviated, others are not.

Author Response

Thank you very much for your comments.

Reviewer 2 Report

This manuscript aims to describe "Efficacy of Endodontic Disinfection Protocols in an E. faecalis Biofilm Model - Using DAPI Staining and SEM". The manuscript is well written and presents an interesting topic, however some minor areas of concern need to be addressed before this manuscript is ready for primetime.

Materials and methods:

Page 3, line 110: Change the uppercase letter F of "Faecalis" to lowercase. In the same line, it is not clear whether the strain is an ATCC line or a clinical bacterium. It must be specified. If it is ATCC, indicate the code. If it is clinical, it must indicate if it was isolated from the environment, oral, endodontic or from another place in the body. Mention the assigned name, the methodology used to identify it as E. faecalis, and mention the bioethics committee certificate that authorized the collection of the sample from humans.

Page 4, line 175: Counting bacetia method may be low described. More details may be given relative to how was calculated the number of bacteria. What is the brand and manufactured compact manual cell counter?

Results:

Figure 3 and Figure 5 appear whit low resolution.  A red line under the word "group 2" should be removed.

While presentation of exact means is useful, inclusion of the standard deviation helps provide useful information to future research based on this work. I suggest including the data in a supplementary table

Discussion

Irrigants possess antibiofilm capabilities. It would have been pertinent to evaluate this property, if there is literature that supports this important property in the field of dentistry where multispecies biofilm plays a fundamental role. In addition to this, discuss the disadvantage of working with a monospecies model that is far from the reality that occurs in the root canal system.

If the monospecie model used the ATCC strain, it should mention the disadvantages of using this type of strain versus clinical strains, specifically isolated from the root canal system.

Author Response

Thank you very much for your comments.

Round 2

Reviewer 1 Report

The authors improve the article after the reviewer's comments.